# Rice Labeling according to Grain Quality Features Using Laser-Induced Breakdown Spectroscopy

**DOI:** 10.3390/foods12020365

**Published:** 2023-01-12

**Authors:** Michael Pérez-Rodríguez, Alberto Mendoza, Lucy T. González, Alan Lima Vieira, Roberto Gerardo Pellerano, José Anchieta Gomes Neto, Edilene Cristina Ferreira

**Affiliations:** 1Tecnologico de Monterrey, Escuela de Ingeniería y Ciencias, Ave. Eugenio Garza Sada 2501, Monterrey 64849, N.L., Mexico; 2Institute of Chemistry, São Paulo State University—UNESP, R. Prof. Francisco Degni 55, Araraquara 14800-900, SP, Brazil; 3Instituto de Química Básica y Aplicada del Nordeste Argentino (IQUIBA-NEA), Consejo Nacional de Investigaciones Científicas y Técnicas (CONICET), Facultad de Ciencias Exactas y Naturales y Agrimensura, Universidad Nacional del Nordeste—UNNE, Ave. Libertad 5470, Corrientes 3400, Argentina

**Keywords:** rice, grain quality features, LIBS, spectral processing, k-nearest neighbors

## Abstract

Rice is an important source of nutrition and energy consumed around the world. Thus, quality inspection is crucial for protecting consumers and increasing the rice’s value in the productive chain. Currently, methods for rice labeling depending on grain quality features are based on image and/or visual inspection. These methods have shown subjectivity and inefficiency for large-scale analyses. Laser-induced breakdown spectroscopy (LIBS) is an analytical technique showing attractive features due to how quick the analysis can be carried out and its capability of providing spectra that are true fingerprints of the sample’s elemental composition. In this work, LIBS performance was evaluated for labeling rice according to grain quality features. The LIBS spectra of samples with their grain quality numerically described as Type 1, 2, and 3 were measured. Several spectral processing methods were evaluated when modeling a k-nearest neighbors (k-NN) classifier. Variable selection was also carried out by principal component analysis (PCA), and then the optimal k-value was selected. The best result was obtained by applying spectrum smoothing followed by normalization by using the first fifteen principal components (PCs) as input variables and k = 9. Under these conditions, the method showed excellent performance, achieving sample classification with 94% overall prediction accuracy. The sensitivities ranged from 90 to 100%, and specificities were in the range of 92–100%. The proposed method has remarkable characteristics, e.g., analytical speed and analysis guided by chemical responses; therefore, the method is not susceptible to subjectivity errors.

## 1. Introduction

Rice (*Oryza sativa* L.) is a staple cereal crop in many countries, and it is an important source of nutrition and energy for the population. Its main constituents are starch and proteins, besides vitamins, fibers, and minerals [1,2]. Rice grain quality is directly reflected in its market value and acceptance by consumers. Nevertheless, the measurement of quality is complex due to the diversity of intrinsic and extrinsic attributes [3]. The main quality parameters are grain yield, cooking properties, nutritional value, texture, grain chalkiness and shape, integer grain percentage, and gelatinization temperature [4,5].

The lack of an international consensus on the quality of rice grain for marketing induced countries to establish their own compliance regulations. In Brazil, rice is classified into groups (shelled or processed), subgroups (natural, parboiled, integral, polished, integral parboiled, and polished parboiled), classes (short, medium, long, long fine, and mixed), and types. The types, also referred to as grain quality, are numerically defined (1 to 5) according to the percentage of defects (foreign matter and impurities, moldy and sour grains, chopped or stained, plastered, or green, red, and yellow) and broken grains. The greater the number of the type, the greater the percentage of defects [6]. Therefore, the type’s number takes into account the quality factors associated with cleanliness, uniformity, sanitary conditions, and product purity, which are trends consistent with international market perceptions [7].

Rice is classified into types according to Brazilian standards by visual inspection and separation. The masses of defects are then determined, and the percentage is estimated, allowing the classification of rice into types [6]. Image analyses of grains have also been widely used to identify defects in rice samples, e.g., grain discoloration, size, and chalkiness [8,9]; external defects [10]; and disease [11]. The challenge of image and/or visual inspection methods is to analyze the grains one by one in an entire batch. These methods of analysis provide analyst fatigue and subjectivity, and they are also slow and inefficient for large-scale analysis. Therefore, the lack of fast, accurate, and robust analytical methods for the classification of rice is still a research gap.

Laser-induced breakdown spectroscopy (LIBS) is an emission spectroscopic technique capable of performing multielement analyses. To achieve this aim, a high energy laser pulse is used to ablate a small mass of a sample, promoting the breakdown of chemical bonds, vaporization, and plasma formation. In the plasma, the species are excited, and their decays provide radiation emissions at specific wavelengths. After separation, the wavelengths are recorded in a spectrum (signal intensities vs. wavelength), which can be used as a sample signature and also as data source for determining a sample’s chemical constituents [12]. LIBS can promote direct and rapid analysis, requiring minimum or no sample preparation, without chemical residues generation [13]. Several authors have demonstrated the LIBS potential to analyze a large number of food matrices [14,15,16,17,18,19,20]. To aid spectra processing and extract relevant spectral features, LIBS data have been commonly allied to pattern recognition.

Within the scope of rice analysis, LIBS methods apply sample authentication according to their geographical origin [15,21,22], and genotypic varieties [23,24] can also be found. However, determining the rice’s quality related to grain defects using the LIBS technique is still inadequate. This approach becomes novel since rice can be labeled based on chemical measurements that are free of subjectivity. 

Considering the above, the novelty of this paper comes from the investigation of LIBS data using k-nearest neighbors (k-NN) and principal component analysis (PCA) modeling for easy rice classification according to grain quality features.

## 2. Materials and Methods

### 2.1. Samples

A total of sixty polished rice samples purchased from local supermarkets in the city of Araraquara (São Paulo State, Brazil) were analyzed. Samples were labeled according to their commercial quality type, namely Type 1, Type 2, and Type 3. The maximum percentage of defects allowed by the Ministry of Agriculture, Livestock and Supply [6] for each polished rice type is shown in Table 1. The studied classes were made up of 25, 20, and 15 rice samples for Type 1, 2, and 3, respectively.

The rice samples were singly homogenized by cryogenic grinding using a cryogenic mill from Spex 6750 (Metuchen, NJ, USA). The grinding program consisted of 2.0 min of pre-freezing, 2.0 min of grinding, and 3.0 min of freezing between the two milling steps. Around 250 mg of each powdered sample was used to prepare pellets by applying 10 tons using a mechanical press (Solab SL—10/15, Piracicaba, Brazil). In this step, 2 pellets were prepared for each sample, resulting in a total of 120 pellets.

### 2.2. LIBS Instrumentation and Measurements

Rice pellets were analyzed using a LIBS system equipped with a 1064 nm Q-switched Nd:YAG laser (Quantel, Big Sky Ultra 50, Bozeman, USA) operating at 50 mJ maximum power energy with a pulse duration that is less than 8 ns, lens for laser focalization with focal distance of 10 cm, and an optical fiber bundle to lead plasma emissions to four spectrometers (Ocean Optics HR2000+, Dunedin, USA), which feature the spectral resolution of 0.1 nm FWHM (full width at half maximum) and cover a spectral range from 200 to 600 nm. The instrumental fixed integration time was 1 ms and the Q-Switch delay was set to 2.5 μs. Additionally, a spark discharge device, previously proposed by Vieira et al. [25] was coupled to the LIBS system to increase the sensitivity of the measures. The output discharge was set to 4.5 kV. The OOLIBS software from Ocean Optics, USA, was used to control the instrument and for data acquisition.

For analysis, rice pellets were placed in LIBS’s sample holder, which was moved in x and y directions relative to each laser pulse with a pulse duration of 20 ns, providing spectra acquisitions corresponding to fifty pulses spread on the pellets’ surface. The interaction laser sample provided a spot diameter of around 300 μm and irradiance in the focal point around 6.2 GW cm^−2^. In addition, a video camera inside the sampling chamber was used for monitoring the analysis.

### 2.3. Chemometric Data Analysis

#### 2.3.1. Preprocessing

Fluctuation effects due to the microheterogeneity of samples, different laser-sample interactions, and laser fluctuations can mask a sample’s spectral features [12]. Therefore, preprocessing the LIBS spectra is commonly used to minimize undesirable fluctuation effects. Aiming to find the most appropriate spectral preprocessing twenty-eight methods were evaluated individually and combined. The evaluated preprocessing methods comprised the following: mean centering, autoscaling, baseline correction by the Whittaker filter, Savitzky–Golay first derivative, normalization by intensity, standard normal variate (SNV), multiplicative scatter correction (MSC), and smoothing. The application of these preprocessing methods introduces significant advantages, such as a reduction in modeling complexity and high variance explanations by using fewer principal components. The preprocessing performance for classification was evaluated by fitting k-NN models, which had all wavelengths of the LIBS spectra in the range of 300–600 nm as input variables. The overall accuracy of k-NN predictions, calculated as the ratio between all correct predictions and total number of examined cases, was used for preprocessing comparisons. 

#### 2.3.2. Variable Selection

As modeling speed and stability can be improved by using the number of PCs instead of the original variables for classification, principal component analysis (PCA) was applied to reduce the dimension of data and to replace the original variables with the principal components (PCs). This procedure also removes correlations and redundant information commonly present in original data [26]. The criteria used to select the number of PCs as input variables was the 100% explanation provided by the accumulated variance.

#### 2.3.3. Model Optimization and Validation

The k-NN algorithm was used for modeling, and the Euclidian distance was used to calculate classification accuracies. This algorithm was selected due to its modeling speed and stability in the prediction’s results even for small sample groups. K-NN is a distance-based model using distance among objects to ascribe one of them to the most common class among the k-nearest neighbors. Considering that the optimization of the k-value is a critical point in the development of k-NN models, this task was carried out. Cross-validation is an efficient method to determine a good k-value by using an independent dataset.

Models were validated by five-fold cross-validation methods, avoiding the occurrence of bias. For this purpose, the spectral data matrix was randomly split into five mutually exclusive subsets, and each classifier was trained and tested five times so that all samples were used to test the fitted models for at least one time. In this validation process, the metrics were calculated as an average of five results. The highest accuracy was used as the selection criteria for optimizing the k-value during the training step. 

#### 2.3.4. Classification Metrics and Software

The sensitivity (correct positive predictions divided by the number of positive cases), specificity (correct negative predictions divided by the number of negative cases), and Kappa statistics were also measured to assess the performance of the fitted models [27]. A detailed description for the Kappa agreement coefficient calculation can be found in McHugh’s study [28]. All calculations were made using R-project software (R Core Team, 2020) version 3.6.3 with caret and chemometric-with-R packages [29].

## 3. Results and Discussion

The LIBS spectra of rice show a large number of emission lines due to the complex composition of the samples. Despite the instrumental range available for data acquisition (200–600 nm), the range of 200–300 nm showed very low intensity signals, and so it was not used. For a better understanding of sample composition, emission lines recorded in the 300–600 nm range were identified. Table 2 lists the main characteristic lines observed in the emission spectra of different rice types.

The qualitative analysis of the emission lines revealed that the rice samples with distinct commercial quality presented similar elemental compositions, since the same lines were identified in all spectra of the three rice types studied. Based on the spectral lines’ intensity, the most abundant elements were N II and Fe I.

Afterwards, aiming to improve the classification’s performance, studies were carried out to evaluate more efficient data preprocessing. The Euclidean distance was used to calculate the distances between the samples, and consequently, the optimal sizes of k neighbors were determined, corresponding to the highest classification accuracy. Although a large k-value is usually more precise because it reduces the mathematical process’s noise, there is no guarantee that overfitting is avoided. The labeling of sample classes was determined by using majority voting. The classification performance of each evaluated preprocessing technique is shown in Figure 1. The results show that the most optimized k-value was five, which provided low accuracy (53%). Normalization, mean centering, and smoothing caused a gradual increase in the number of true positives, while the remaining preprocessing favored the number of true negatives for predictions. The levels of correct classifications ranged from 64.7 to 82.3%. The better-fit model was obtained using the data preprocessed by smoothing followed by normalization, which achieved the highest prediction accuracy (82.3%), with k = 11. This processing enhanced 1.7-fold the model’s performance compared to raw data; therefore, it was selected for the next studies.

Despite the good classification results described above, the model’s performance may be affected by inputting a high number of variables [30]. In the LIBS spectra, 5384 wavelengths were recorded in the used spectral range. Usually, complete data contain redundant/collinear information, which can hinder successful classification. Therefore, the PCA of the preprocessed spectral data was carried out with the aim to compress the original variables, replacing them with the corresponding PCs. 

The discrimination of rice classes was clearly observed by the score plot, as shown in Figure 2, where PC1 + PC2 represents 84.7% of total data variance. The scores in Type 3 showed a greater dispersion compared to the other classes. This can occur due to its higher content of defective grains comprising greater variability in the percentage of individual defects, which show different chemical compositions detected by LIBS. Although the qualitative analysis of rice fingerprints indicated a similar elemental composition for different commercial quality types (Table 2), PCA was apparently capable of detecting quantitative differences among the samples, which enabled the successful discrimination of rice classes. Therefore, the new variables (PCs) were used to fit the k-NN model. As total data variance (100%) was achieved by the contribution of the first fifteen PCs, all these PCs were selected as input variables. 

Figure 3 shows the overall accuracy determined by cross-validation during the training step, in function of k. The highest accuracy associated with the smallest standard deviation was obtained with k = 9. The precision of the predictions for each evaluated neighbor’s number, expressed as the relative standard deviation (%RSD), was within the range of 2.4–9.1%. 

The classification results obtained for the best fitted model are summarized in Table 3. The correct predictions rates per class were 100% for Type 1, 92% for Type 2, and 95% for Type 3. These results indicate that Type 1 can be successfully identified from the remaining classes, probably due to the little changes in the proportionality among its elementary components detected by LIBS. Generally, the samples from Type 2 and Type 3 can be discriminated against each other with 100% specificity and sensitivity, respectively. The overall correct classification rate was 94%, and this is considered impressive for distinguishing among high-quality rice and inferior-quality rice. 

The reliability of the findings reported here was verified by calculating the kappa statistic (Kappa = 0.91), which indicates an almost perfect agreement level among the predicted quality types and the real quality types. of the use of PCs as input variables allowed increasing k-NN performance by reducing modeling complexity. Improvements in the correct classification rates by 2.0-fold allowed a reduction in operation times from 3.24 to 0.35 min, which contributed to modeling stability and measurement reproducibility. The developed method is clean, fast, accurate, robust, and guided by chemical responses; therefore, it is not susceptible to subjectivity errors.

## 4. Conclusions

The association between the LIBS analytical technique and k-NN algorithm was evaluated for rice classification according to grain quality features. The results demonstrate the great potential of this combination for replacing the current methods. The proper spectral processing methods and selection of variables by PCA were fundamental for improving the classifier’s performance, allowing it to achieve 94% accuracy in the overall predictions. The developed method may bring innovation to the rice value chain, especially because it is free of subjectivity, simple, fast, sensitive, specific, accurate, and eco-friendly.

## Figures and Tables

**Figure 1 foods-12-00365-f001:**
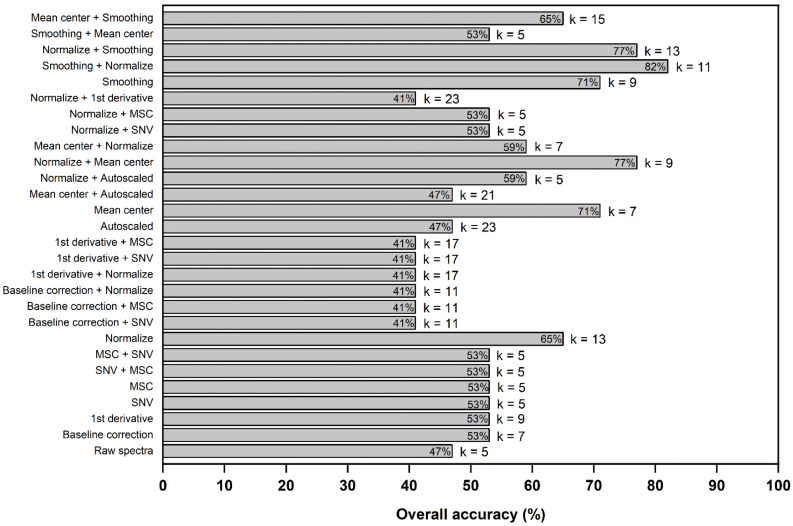
Evaluation of different spectral preprocessing for rice classification by k-NN algorithms.

**Figure 2 foods-12-00365-f002:**
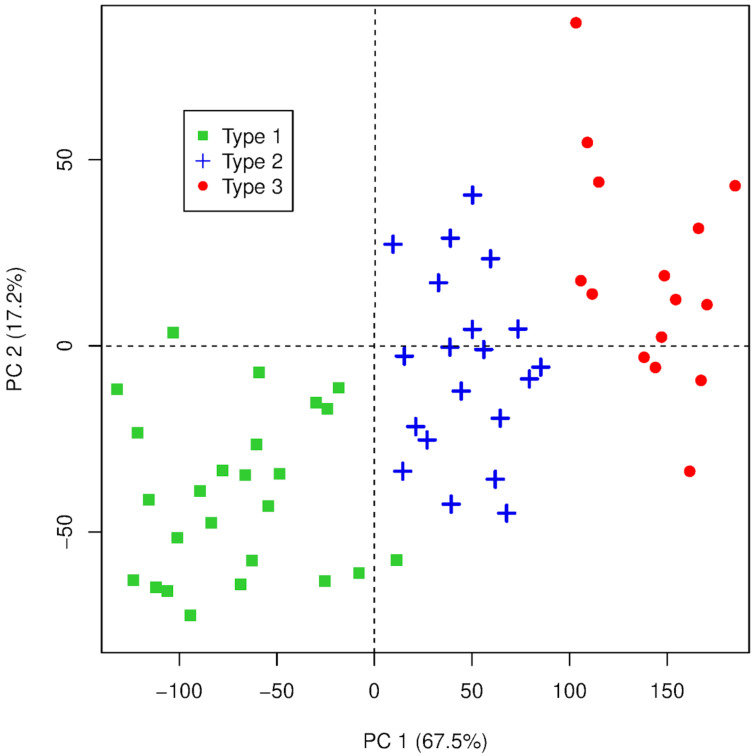
Two-dimensional representation on the PC1 and PC2 scores.

**Figure 3 foods-12-00365-f003:**
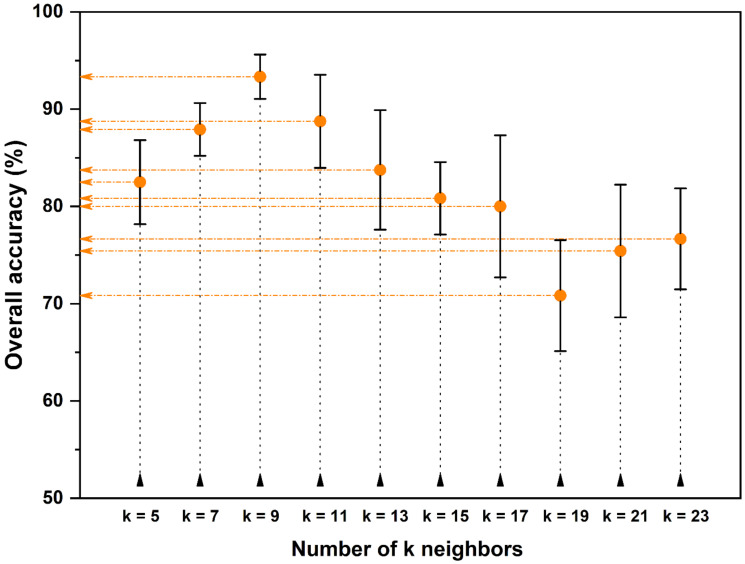
Results of k-value optimization. Error bars represent the standard deviations of five training prediction measurements.

**Table 1 foods-12-00365-t001:** Maximum tolerance limits of defects established for classifying polished processed rice according to the commercial quality type.

Quality Type	General Defects (%) *					
	Moldy and Sour	Chopped or Stained	Plastered or Green	Brindle	Yellow	Broken
1	0.15	1.75	2.00	1.00	0.50	7.00
2	0.30	3.00	4.00	1.50	1.00	14.00
3	0.50	4.50	6.00	2.00	2.00	23.00

* Expressed in weight percent.

**Table 2 foods-12-00365-t002:** Emission lines of rice spectrum.

Element *	Registered Spectral Lines (nm)
C II	426.7
Ca I	397.4
Ca II	393.4
CN	387.1, 388.3
Fe I	302.3, 304.6, 305.9, 326.0, 326.5, 331.9, 335.3, 336.7, 337.4, 339.0, 340.7, 343.7, 347.1, 371.2, 372.7, 375.4, 375.9, 377.0, 383.9, 395.6, 396.9, 397.4, 407.6, 409.8, 410.4, 426.7, 435.0, 444.7, 459.2, 459.6, 460.2, 464.3, 466.2, 470.0, 470.6, 478.9, 553.4, 568.6, 571.0
Fe II	371.2, 375.9, 441.6
Mg I	383.2, 383.9, 518.3
N II	332.9, 399.6, 460.8, 461.5, 462.2, 463.1, 464.3, 478.1, 480.4, 499.5, 504.6, 549.5, 553.4, 566.6, 567.9, 574.7, 593.2, 594.1
Na I	589.0, 589.5
O II	313.5, 374.9, 412.0

* I: atomic spectral lines; II: ionic spectral lines.

**Table 3 foods-12-00365-t003:** Classification metrics of optimized k-NN model.

Grain Quality	Sensitivity (%)	Specificity (%)	Accuracy (%)
Type 1	100	100	100
Type 2	83	100	92
Type 3	100	90	95
Overall			94

## Data Availability

The data generated during the current study are available from the corresponding author upon reasonable request.

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
