# Peer review of "Rice Labeling according to Grain Quality Features Using Laser-Induced Breakdown Spectroscopy"

_foods, 2023, doi:10.3390/foods12020365_

Round 1

Reviewer 1 Report

The authors propose an efficient method for evaluation of rice labeling, according to grain quality features using Laser induced breakdown spectroscopy (LIBS). In this work, LIBS performance was evaluated for labeling rice. LIBS spectra of samples having their grain quality numerically described as type 1, 2, and 3 were measured. Several spectral processing methods were evaluated in modeling a k-nearest neighbors (k-NN) classifier. Variable selection was also carried out by principal component analysis (PCA) and then optimal k-value was selected. The best result was obtained with spectrum smoothing followed by the normalization, using the first fifteen principal components (PCs) as input variables and k=9. Under these conditions, the method showed an excellent performance achieving samples classification with 94% of overall prediction accuracy. The sensitivities ranged from 90 to 100% and specificities were in the range 92–100%. The proposed method has remarkable characteristics, e.g., analytical speed and analysis guided by chemical responses, therefore not being susceptible to errors of subjectivity. This work is valuable for rice breeding community. However, some minor revisions should be made that are given below:

 Line 74: Replace “samples” with “sample”

Line 127: Remove the word “to”

Line 152- 153: Rephrase the sentence

Line 170: Replace “carried” with “carried out”

Line 201: Rephrase the sentence

Author Response

Response to Reviewer 1 Comments

The authors propose an efficient method for evaluation of rice labeling, according to grain quality features using Laser induced breakdown spectroscopy (LIBS). In this work, LIBS performance was evaluated for labeling rice. LIBS spectra of samples having their grain quality numerically described as type 1, 2, and 3 were measured. Several spectral processing methods were evaluated in modeling a k-nearest neighbors (k-NN) classifier. Variable selection was also carried out by principal component analysis (PCA) and then optimal k-value was selected. The best result was obtained with spectrum smoothing followed by the normalization, using the first fifteen principal components (PCs) as input variables and k=9. Under these conditions, the method showed an excellent performance achieving samples classification with 94% of overall prediction accuracy. The sensitivities ranged from 90 to 100% and specificities were in the range 92–100%. The proposed method has remarkable characteristics, e.g., analytical speed and analysis guided by chemical responses, therefore not being susceptible to errors of subjectivity. This work is valuable for rice breeding community. However, some minor revisions should be made that are given below: 

Response: Dear Reviewer, thank you for the time you spent in the analysis of our manuscript and your suggestions. Your comments were all valuable and helpful for revising and improving our paper quality. We uploaded the revised version of the manuscript with all the changes marked up using the “Track Changes” function.

Point 1: Line 74: Replace “samples” with “sample”
Response 1: The suggestion was accepted.

Point 2: Line 127: Remove the word “to”
Response 2: The suggestion was accepted.

Point 3: Line 152- 153: Rephrase the sentence
Response 3: The suggestion was accepted.

Point 4: Line 170: Replace “carried” with “carried out”
Response 4: The suggestion was accepted.

Point 5: Line 201: Rephrase the sentence
Response 5: The suggestion was accepted.

We hope your comments have been fully addressed. If otherwise, we are happy to provide further information.

Yours sincerely,
Authors

Reviewer 2 Report

all comments in the pdf file attached

Author Response

Response to Reviewer 2 Comments

All comments in the pdf file attached

Response: Dear Reviewer, thank you for the time you spent in the analysis of our manuscript and your suggestions. Your comments were all valuable and helpful for revising and improving our paper quality. We uploaded the revised version of the manuscript with all the changes marked up using the “Track Changes” function.

We hope your comments have been fully addressed. If otherwise, we are happy to provide further information.

Yours sincerely,
Authors

Reviewer 3 Report

This work aimed to gain a speedy and unbiased technique to classify rice quality for trading by using laser-induced breakdown spectroscopy (LIBS). Attempt was made by comparing LIPBS spectra from 120 samples, selected from 3 sets of rice quality, as defined by Brazilian standards. Data were carefully processed to reduce instrumental errors to finally obtain k-NN and PCA modelling, including analyses the sensitivity, specificity, and accuracy of the technique.    

Authors stated clear introduction of LIPS applications in differentiating minute amount of chemicals in biological materials, such as rice grains from different growing locations and varieties, but the rice quality had not been included earlier. The objective of this work, therefore, to investigate whether LIPS can be used for differentiating rice quality. Reasonable experimental design and analysis techniques are comprehensible and resulted in possible application of LIBS for rice labelling. However, there are some points that could be added for readers who are interested, but not familiar in the Brazilian rice standard and analysis techniques for LIPS.

There are 3 points that should be added, if possible:

1) Table 1 shows descriptive defects of rice according to Brazilian standard, in which the descriptions could be different from that of the countries from other regions. If pictures can be added (even in black and white), it would really help.

2) Although 3 different types should be enough for PCA analysis, why other 2 types were left to complete 5 types of Brazilian standard.

3) Section 2.3.3 line 141-142, please authors suggest which package software was used to operate data for k-NN algorithm and Euclidian distance. If they were developed by authors, please stated.

(optional) adding a brief concept-workflow could help easier reading.  

Author Response

Response to Reviewer 3 Comments

This work aimed to gain a speedy and unbiased technique to classify rice quality for trading by using laser-induced breakdown spectroscopy (LIBS). Attempt was made by comparing LIPBS spectra from 120 samples, selected from 3 sets of rice quality, as defined by Brazilian standards. Data were carefully processed to reduce instrumental errors to finally obtain k-NN and PCA modelling, including analyses the sensitivity, specificity, and accuracy of the technique.    
Authors stated clear introduction of LIPS applications in differentiating minute amount of chemicals in biological materials, such as rice grains from different growing locations and varieties, but the rice quality had not been included earlier. The objective of this work, therefore, to investigate whether LIPS can be used for differentiating rice quality. Reasonable experimental design and analysis techniques are comprehensible and resulted in possible application of LIBS for rice labelling. However, there are some points that could be added for readers who are interested, but not familiar in the Brazilian rice standard and analysis techniques for LIPS. 
There are 3 points that should be added, if possible:

Response: Dear Reviewer, thank you for the time you spent in the analysis of our manuscript and your suggestions. Your comments were all valuable and helpful for revising and improving our paper quality. We uploaded the revised version of the manuscript with all the changes marked up using the “Track Changes” function.

Point 1: Table 1 shows descriptive defects of rice according to Brazilian standard, in which the descriptions could be different from that of the countries from other regions. If pictures can be added (even in black and white), it would really help.
Response 1: That's right, Table 1 summarizes the general defects of rice according to the types of commercial quality based on the Brazilian standard. It is well known that each country or region establishes quality control standards according to the quality parameters of each particular food. Unfortunately, images of the defects of the rice grains considered in this work are not currently available.

Point 2: Although 3 different types should be enough for PCA analysis, why other 2 types were left to complete 5 types of Brazilian standard.
Response 2: The Brazilian standard contemplates 5 types of rice quality according to the percentual of grain defects. However, in this research only the first three types of rice quality were considered, since a insufficient number of rice samples was obtained for the other two types of commercial quality, which could ruin the development of a reliable rice quality prediction model. Furthermore, the first three types of rice quality (1, 2 and 3) are the most commonly found in the Brazilian rice market.

Point 3: Section 2.3.3 line 141-142, please authors suggest which package software was used to operate data for k-NN algorithm and Euclidian distance. If they were developed by authors, please stated.
(optional) adding a brief concept-workflow could help easier reading.  
Response 3: The k-NN modeling was made using R-project software version 3.6.3 (R Core Team, 2017) with caret and chemometric-with-R packages. This information was included in Section 2.3.4. The packages used belong to the R-project software package.

We hope your comments have been fully addressed. If otherwise, we are happy to provide further information.

Yours sincerely,
Authors
